# Identification of suitable target/E3 ligase pairs for PROTAC development using a rapamycin-induced proximity assay (RiPA)

Bikash Adhikari[1], Katharina Schneider[1], Mathias Diebold[1,2], Christoph Sotriffer[2], Elmar Wolf[1]*

[1]Institute of Biochemistry, University of Kiel, Kiel, Germany; [2]Institut für Pharmazie und Lebensmittelchemie, University of Würzburg, Würzburg, Germany

## eLife Assessment

The study describes a **valuable** new technology in the field of targeted protein degradation that allows identification of E3-ubiquitin ligases that target a protein of interest. The presented data are **convincing**, however, additional work will be needed to optimize for high-throughput evaluation. This technology will therefore serve the community in the initial stages of developing targeted protein degraders.

**Abstract** The development of proteolysis targeting chimeras (PROTACs), which induce the degradation of target proteins by bringing them into proximity with cellular E3 ubiquitin ligases, has revolutionized drug development. While the human genome encodes more than 600 different E3 ligases, current PROTACs use only a handful of them, drastically limiting their full potential. Furthermore, many PROTAC development campaigns fail because the selected E3 ligase candidates are unable to induce degradation of the particular target of interest. As more and more ligands for novel E3 ligases are discovered, the chemical effort to identify the best E3 ligase for a given target is exploding. Therefore, a genetic system to identify degradation-causing E3 ligases and suitable target/E3 ligase pairs is urgently needed. Here, we used the well-established dimerization of the FKBP12 protein and FRB domain by rapamycin to bring the target protein WDR5 into proximity with candidate E3 ligases. Strikingly, this rapamycin-induced proximity assay (RiPA) revealed that VHL, but not Cereblon, is able to induce WDR5 degradation - a finding previously made by PROTACs, demonstrating its predictive power. By optimizing the steric arrangement of all components and fusing the target protein with a minimal luciferase, RiPA can identify the ideal E3 for any target protein of interest in living cells, significantly reducing and focusing the chemical effort in the early stages of PROTAC development.

## Introduction

New technologies such as genome sequencing and genetic screens have led to a drastic increase of knowledge about disease-causing proteins and potential therapeutic targets. However, the therapeutic exploitation of this knowledge was often limited by the fact that only a minority of the newly identified targets could be blocked by conventional, i.e., monovalent inhibitors. Thus, most of the existing therapies are so far directed against proteins with enzymatic activity, leaving about 80% of the intracellular proteome undruggable (*Oprea et al., 2018*). This limitation is largely eliminated by the concept of proteolysis targeting chimeras (PROTACs). PROTACs are bifunctional small molecules which bind to their target and a cellular E3 ligase and hence induce target ubiquitylation

*For correspondence:
elmar.wolf@biochem.uni-kiel.de

and degradation (*Bondeson et al., 2015*; *Ishida and Ciulli, 2021*; *Winter et al., 2015*). In addition to the ability of PROTACs to block the primary function of many intracellular targets, proteolytic degradation of disease-causing proteins offers further advantages over conventional inhibition. First, complete degradation results in the inactivation of any protein function, such as all enzymatic activities in multi-enzyme proteins or non-enzymatic activities in enzymes with additional scaffolding functions. Second, because PROTACs remain active after once having induced the degradation of their target (=catalytic mode of action), they act more persistently and at lower drug concentrations, making them more specific than classical inhibitors (=occupancy-based mode of action) (*Farnaby et al., 2021*). Third, since the transient formation of the ternary complex of target, PROTAC, and E3 ligase is sufficient to ubiquitylate the target, PROTACs are less susceptible to the formation of resistance-mediating mutations that reduce the affinity of the target for the drug (*Posternak et al., 2020*). Therefore, despite its novelty, the PROTAC concept has transformed the development of targeted therapies, and numerous PROTACs are already in clinical development with the first cases of successful clinical trials recently being reported (*Chirnomas et al., 2023*; *Montoya et al., 2024*).

Despite the revolutionary potential of PROTACs, their development is also fraught with difficulties. Probably the most serious limitation is the current restricted coverage of E3 ligases. While the human genome encodes for more than 600 E3 ligases, only a few are used to develop PROTACs (*Ishida and Ciulli, 2021*), and most PROTACs are based on only two of them, Cereblon (CRBN) and the von Hippel-Lindau tumor suppressor (VHL). VHL- and CRBN-based PROTACs are potent in inducing degradation of their targets but also have drawbacks. For example, both E3 ligases are commonly expressed in human tissues preventing their use for tissue-specific degradation. In addition, both are not essential for many cells in our body, which could lead to the rapid development of resistance if the respective E3 ligase machinery is silenced, e.g., in the context of an oncology application. Intensive efforts are therefore being made to identify suitable ligands for additional E3 ligases which overcome these shortcomings.

A second major limitation is that the development of PROTACs is largely empirical, with few generally applicable design rules (*Ward et al., 2023*). As a result, many PROTAC development campaigns fail due to the inability to identify PROTACs that efficiently induce target degradation, and in most cases, these results are not published. A possible reason for this frequent failure could be the incompatibility of a particular E3 ligase with a given target. In fact, it has been observed that promiscuous kinase inhibitors exhibit very distinct degradation profiles when used as ligands for PROTACs, demonstrating that the efficacy of PROTAC-mediated degradation is not determined solely by the sheer affinity of the target and E3 ligand but rather by efficient ternary complex formation (*Donovan et al., 2020*). We made similar findings on PROTACs we developed for Aurora A kinase. While the kinase inhibitor we used for PROTAC development also binds and inhibits the homologous enzyme Aurora B, our Cereblon-based PROTACs show remarkable specificity toward Aurora A (*Adhikari et al., 2020*). We observed that certain amino acids on the putative Cereblon interaction surface of Aurora A are not conserved in Aurora B, and that mutation of these residues decreases PROTAC-mediated degradation of Aurora A, indicating that ternary complex formation is supported by direct interactions between Aurora A and Cereblon. In line with our finding, the Ciulli lab demonstrated that cooperativity in ternary complex formation is relevant for PROTAC efficacy (*Roy et al., 2019*; *Gadd et al., 2017*). Taken together, these results suggest that certain E3 ligases are better suited for specific targets than others.

The problem that only certain E3 ligases can efficiently degrade a particular target protein is exacerbated by the rapid discovery of new ligands for additional E3 ligases (such as IAPs [*Itoh et al., 2010*; *Demizu et al., 2016*], DCAF11 [*Zhang et al., 2021*], DCAF15 [*Han et al., 2017*; *Du et al., 2019*], DCAF16 [*Zhang et al., 2019*], RNF4 [*Ward et al., 2019*], RNF114 [*Spradlin et al., 2019*; *Luo et al., 2021*], AhR [*Ohoka et al., 2019*], FEM1B [*Henning et al., 2022*], and KEAP1 [*Konopleva et al., 2002*; *Wei et al., 2021*; *Pei et al., 2023*]). In fact, the availability of new E3 ligase ligands will increase the failure rate of PROTAC development campaigns for reasons more trivial than the inability to efficiently form ternary complexes (*Figure 1A*). Since these E3 ligases will be much less characterized than Cereblon and VHL, it may be unknown whether the target exposes a lysine residue within reach of the E3 ligase, or whether these E3 ligases are even able to attach degradative ubiquitin chains (i.e. K48, K11) or are located in the same cellular compartment as the target.

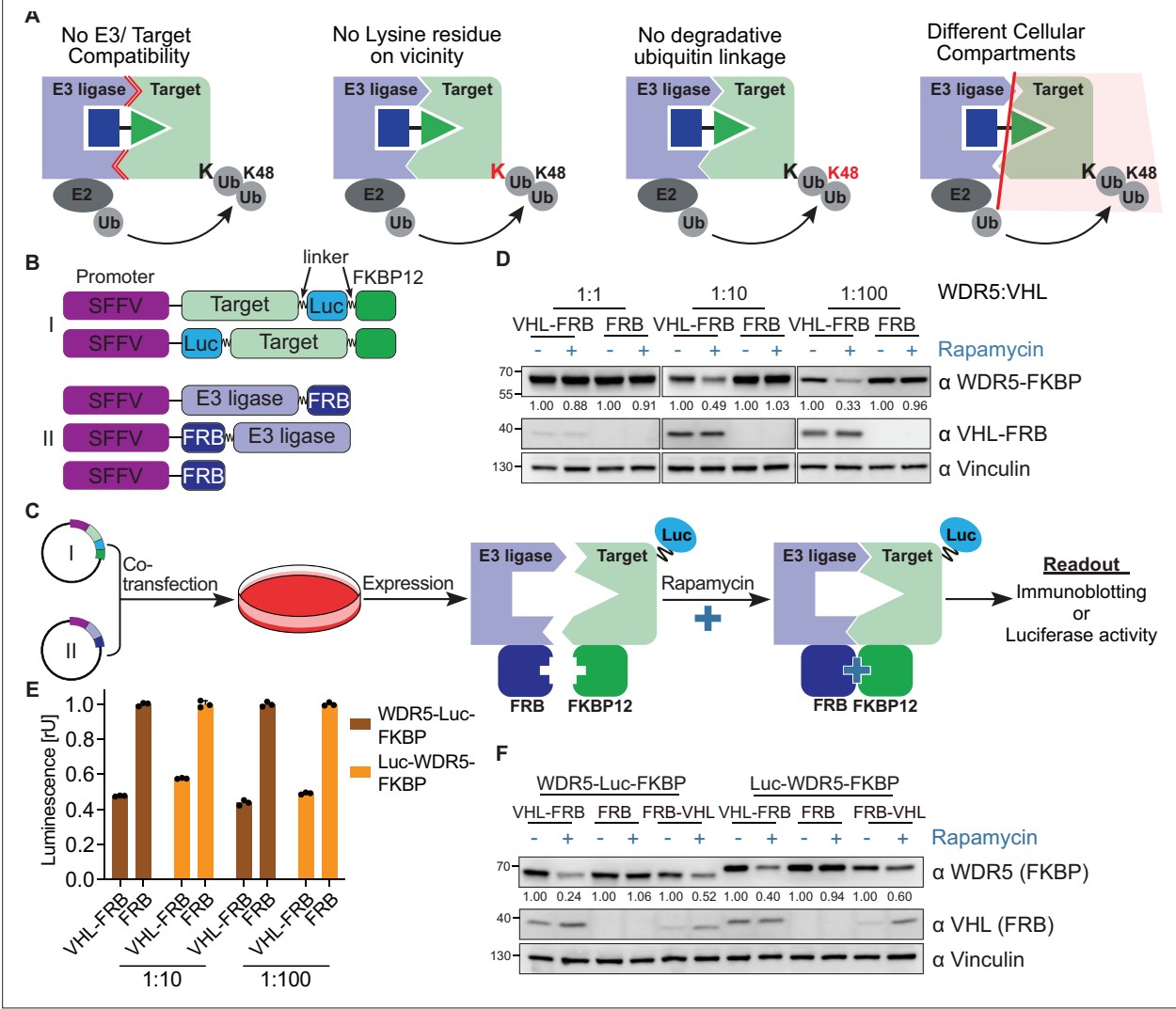

**Figure 1.** Rapamycin-induced proximity assay (RiPA) induces quantifiable degradation of target proteins. (**A**) Schematic illustration of scenarios where proteolysis targeting chimera (PROTAC) could not induce degradation of a target protein. (**B**) Scheme of target protein (**I**) and E3 ligase or control (**II**) constructs used in RiPA. The linker indicated is 2x GSSG in all constructs unless stated otherwise. (**C**) Schematic describing the RiPA experimental protocol. (**D**) Immunoblot of WDR5 and VHL. HEK293 cells were co-transfected with WDR5-Luc-FKBP12 and VHL-FRB or FRB in the indicated ratio and treated with 100 nM rapamycin or vehicle for 6 hr after ~24 hr of expression. Vinculin was used as a loading control (as in all other immunoblotting experiments). (**E**) WDR5 levels based on luciferase measurements. HEK293 cells were co-transfected with WDR5-Luc-FKBP12 or Luc-WDR5-FKBP12 and VHL-FRB or FRB constructs in the indicated ratio, expressed for ~24 hr, and treated with rapamycin overnight. Bars represent mean ± s.d. of n=3 replicates. (**F**) Immunoblot of WDR5 and VHL. HEK293 cells were co-transfected with a combination of WDR5-Luc-FKBP12 or Luc-WDR5-FKBP12 and VHL-FRB or FRB-VHL or FRB in the ratio of 1:10, expressed for ~24 hr and treated with rapamycin overnight. WDR5 and VHL fusion proteins tagged at the N- and C-terminal show different migration behaviors despite having same molecular weight.

The online version of this article includes the following source data and figure supplement(s) for figure 1:

**Source data 1.** PDF file containing original western blot for *Figure 1D and F*, indicating the relevant bands.

**Source data 2.** Original files for western blot analysis displayed in *Figure 1D and F*.

**Figure supplement 1.** Rapamycin-induced proximity assay (RiPA) induces quantifiable degradation of target proteins.

We therefore see an urgent need for genetic assays that allow us to identify suitable target/E3 ligase pairs. This assay should not only be target-specific but also applicable to different cellular systems. In addition, it should be scalable to ideally cover a high number of E3 ligase candidates while remaining easy to use. We decided to exploit the widely used rapamycin-dimerization system to test whether induced cellular proximity of a specific target/E3 ligase pair induces target degradation. Our system correctly predicted PROTAC-mediated degradation of candidate target proteins and allows

accurate, time-resolved estimation of target degradation by measuring luciferase activity in living cells, and will likely streamline medicinal chemistry efforts during PROTAC development.

## Results

### RiPA induces quantifiable degradation of target proteins

To induce artificial proximity between target proteins and cellular E3 ligases, we exploited the rapamycin-induced dimerization of the proteins, FK506 binding protein (FKBP12), and the FKBP12-rapamycin binding (FRB) domain of FKBP12-rapamycin associated protein (mTOR) (*Figure 1—figure supplement 1A*; *Choi et al., 1996*). We cloned FRB and FKBP12 into lentiviral vector systems that allow both transient transfection and stable cell line generation by viral transduction. We also included a multiple cloning site (MCS) for easy target protein cloning and enabled robust expression using the spleen focus forming virus (SFFV) promoter (*Figure 1B*).

To test whether this rapamycin-induced dimerization assay (RiPA) could induce target degradation, we inserted the WD repeat-containing protein 5 (WDR5) and the von Hippel-Lindau tumor suppressor (VHL) into the FKBP12 and FRB-containing plasmids, respectively. We chose this target/E3 ligase pair because we (*Dölle et al., 2021*) and others *Yu et al., 2023*; *Yu et al., 2021* have previously demonstrated robust degradation of WDR5 by PROTACs harnessing VHL. We started to optimize the RiPA system by transfecting different amounts of the WDR5 and VHL-encoding plasmids into HEK293 cells and incubating them with 0.1 μM rapamycin for 6 hr (*Figure 1C*). Immunoblotting showed that rapamycin induced a marked reduction in WDR5-FKBP12 when a 10- or 100-fold excess of VHL-FRB was expressed, but not when the target protein and E3 ligase were transfected at equimolar ratios (*Figure 1D*). FRB alone did not decrease WDR5-FKBP12 levels, demonstrating that VHL causes degradation of WDR5-FKBP12 in response to rapamycin. For a more quantitative and convenient readout of target protein levels, we fused a *Oplophorus gracilirostris*-based minimal luciferase (*Hall et al., 2012*) to either the N- (Luc-WDR5-FKBP12) or C-terminus (WDR5-Luc-FKBP12) of WDR5 (*Figure 1—figure supplement 1B*) and expressed them in HEK293 cells. Both conditions allowed robust measurement of luciferase activity, demonstrating that terminal and internal tagging are compatible with luciferase activity. Strikingly, rapamycin reduced luciferase activity by 50.2% (±5.8 %) when expressed together with a 10- or 100-fold excess of VHL-FRB (*Figure 1E*). Immunoblotting confirmed the robust degradation of luciferase and FKBP12-tagged WDR5 by VHL (*Figure 1F*).

### RiPA correctly predicts suitability of E3 ligases for WDR5 PROTACs

We next wanted to evaluate whether the RiPA system has predictive power, i.e., whether it can discriminate between suitable and unsuitable target/E3 ligase pairs, or whether any E3 ligase can lead to target degradation after induced proximity. To this end, we expressed the E3 ligase Cereblon in fusion with FRB (CRBN-FRB) under the same conditions and compared its ability to degrade WDR5 with VHL-FRB. Both immunoblotting and luciferase assays again revealed a robust and time-dependent degradation of WDR5 by VHL-FRB but not by CRBN-FRB (*Figure 2A and B*). This is relevant because several studies with PROTACs indicate efficient degradation of WDR5 by VHL but not by CRBN (*Dölle et al., 2021*; *Yu et al., 2021*). In contrast, CRBN-based PROTACs have been shown to efficiently degrade the mitotic kinase Aurora A (*Donovan et al., 2020*; *Adhikari et al., 2020*; *Wang et al., 2021*; *Rishfi et al., 2023*; *Liu et al., 2022*). We therefore tested whether VHL and CRBN are able to degrade Aurora A in the RiPA system. In contrast to WDR5, both E3 ligases induced a robust degradation of Aurora A as shown by immunoblotting (*Figure 2C*) and luciferase activity (*Figure 2D*).

While these results suggest that the respective E3 ligase ubiquitylates the target protein (WDR5 or Aurora A), it cannot be excluded that luciferase or FKBP12 could also serve as a substrate for ubiquitylation. In fact, luciferase and FKBP12 proteins contain 7 and 8 lysine residues, respectively. Therefore, we analyzed the available crystal structures of both proteins and found that all 15 lysine residues are located on the protein surfaces accessible for post-translational modifications (*Figure 2E*). To exclude that ubiquitylation of luciferase or FKBP12 interfered with the RiPA results, we converted all lysine residues to arginine and expressed WDR5 together with these lysine-less (Kless) versions of luciferase and FKBP12. While lysine substitution reduced luciferase activity by 35.4%, the measurement of luciferase activity was still reliable (*Figure 2F*). Strikingly, rapamycin-induced proximity to VHL under these conditions induced WDR5 degradation comparable to lysine-containing protein tags

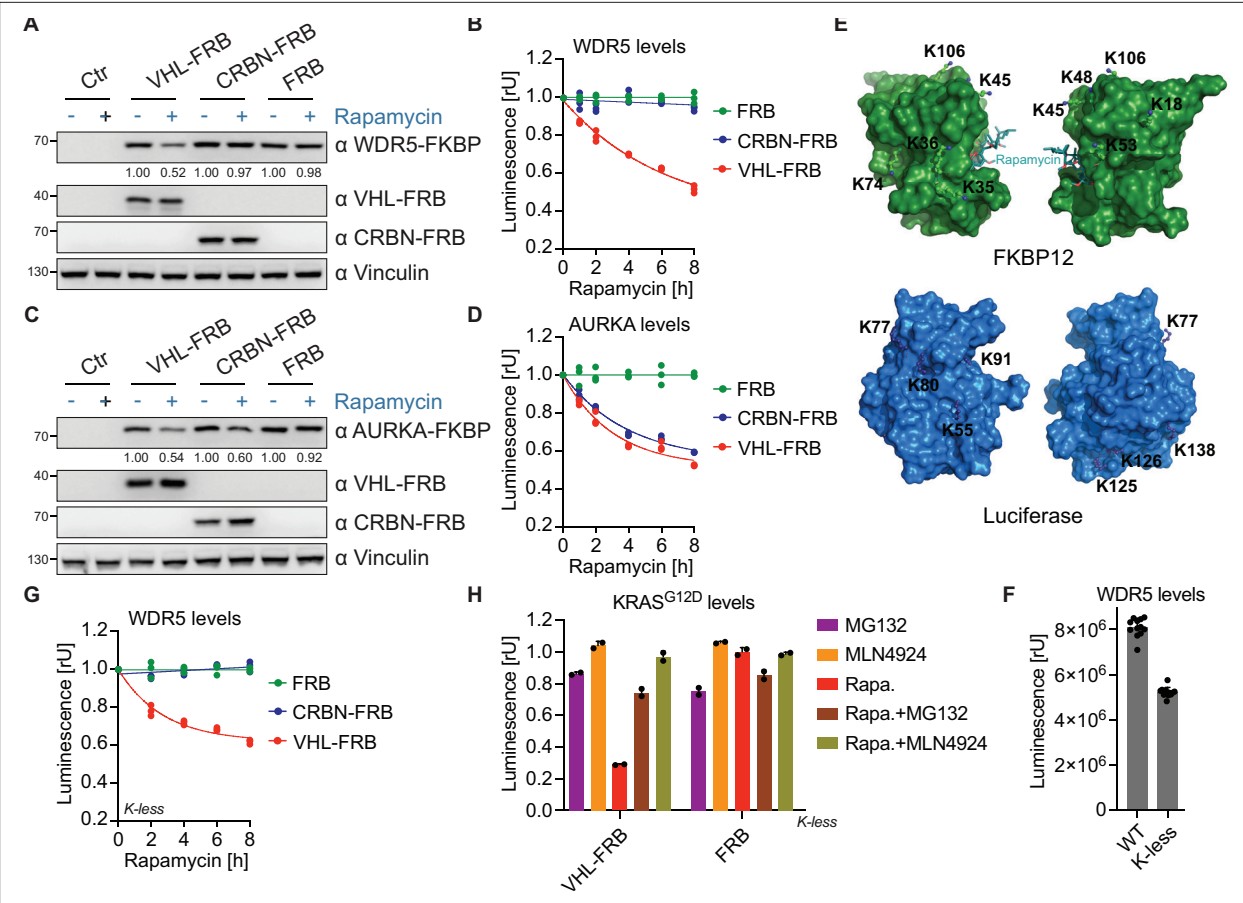

**Figure 2.** Rapamycin-induced proximity assay (RiPA) correctly predicts suitability of E3 ligases for WDR5 proteolysis targeting chimeras (PROTACs). (**A**) Immunoblot of WDR5, VHL, and CRBN. HEK293 cells were transfected with WDR5-Luc-FKBP12 and VHL-FRB or CRBN-FRB or FRB in a 1:10 ratio, expressed for 24 hr, and treated with 10 nM rapamycin for 6 hr. (**B**) WDR5 levels based on luciferase measurement. Luminescence was measured in HEK293 cells as described in (**A**) after 10 nM rapamycin treatment at specified time points. Data are shown as individual values (n=3 replicates), with a curve fitted to the mean across replicates. (**C**) Immunoblot of AURKA, VHL, and CRBN. HEK293 cells were transfected with AURKA-Luc-FKBP12 and VHL-FRB or CRBN-FRB or FRB in a 1:10 ratio and treated with 10 nM rapamycin for 6 hr. (**D**) AURKA levels based on luciferase measurement. Luminescence was measured in HEK293 cells as described in (**C**) after 10 nM rapamycin treatment at indicated time points. Data are shown as individual values (n=3 replicates), with a curve fitted to the mean across replicates. (**E**) Structure of FKBP12 and luciferase. Molecular surface representation of FKBP12 (top) and luciferase (bottom) showing lysine residues on their surface. The lysine residues are labeled and two sides for each protein are shown. (**F**) WDR5 levels based on luciferase measurement. HEK293 cells were co-transfected with either WDR5-Luc-FKBP12 (WT) or WDR5-Luc-FKBP12 construct where all lysine residues on Luc and FKBP12 were mutated to arginine (Kless) and FRB, expressed for ~24 hr and luminescence measured. Bars represent mean ± s.d. of n=12 replicates. (**G**) WDR5 levels based on luciferase measurement. Luminescence was measured in HEK293 cells expressing WDR5-Luc-FKBP12(Kless) and VHL-FRB or CRBN-FRB or FRB after 10 nM rapamycin treatment at specified time points. Data are shown as individual values (n=3 replicates), with a curve fitted to the mean across replicates. (**H**) KRAS$^{G12D}$ levels based on luciferase measurements. HEK293 cells were co-transfected with KRAS$^{G12D}$-Luc-FKBP12(Kless) and VHL-FRB or FRB constructs, expressed for ~24 hr, and treated with 10 nM rapamycin (Rapa.) in the presence or absence of 10 µM MG132 and 5 µM MLN4924 for 8 hr. Bars represent mean ± s.d. of n=2 replicates.

The online version of this article includes the following source data and figure supplement(s) for figure 2:

**Source data 1.** PDF file containing original western blot for **Figure 2A and C**, indicating the relevant bands.

**Source data 2.** Original files for western blot analysis displayed in **Figure 2A and C**.

**Figure supplement 1.** Rapamycin-induced proximity assay (RiPA) predicts suitability of E3 ligases for proteolysis targeting chimeras (PROTACs) against various targets.

**Figure supplement 1—source data 1.** PDF file containing original western blot for **Figure 1A**, indicating the relevant bands.

**Figure supplement 1—source data 2.** Original files for western blot analysis displayed in **Figure 1A**.

(*Figure 2G*, *Figure 2—figure supplement 1A*), demonstrating that VHL induces direct ubiquitylation of WDR5.

To further evaluate our system, we cloned Aurora B, another member of the Aurora kinase family, as well as the oncogenic KRAS into our Kless luciferase-FKBP12 construct. While both VHL and CRBN exhibited comparable degradation efficiency against Aurora A, VHL was notably more effective than CRBN in degrading Aurora B (*Figure 2—figure supplement 1B*). Previous work by Zeng et al. demonstrated that CRBN-harnessing degrader targeting oncogenic KRAS$^{G12C}$ using a GFP-tagged KRAS$^{G12C}$ reporter system failed to degrade the endogenous KRAS$^{G12C}$, suggesting that the degrader primarily ubiquitinated the GFP tag rather than the KRAS$^{G12C}$ itself (*Zeng et al., 2020*). Other studies have consistently reported that VHL-recruiting degraders efficiently degrade KRAS$^{G12D}$ and other KRAS mutants compared to CRBN-recruiting degraders (*Zhou et al., 2024*; *Lim et al., 2021*; *Bond et al., 2020*; *Yang et al., 2022*; *Zhang et al., 2023*; *Yang et al., 2023*). Consistent with these findings, our system demonstrated efficient degradation of KRAS$^{G12D}$ by VHL-FRB. In contrast, rapamycin-induced dimerization with CRBN-FRB only lead to minor target removal (*Figure 2—figure supplement 1C*). Next, to validate the system and confirm the mechanism of action, we conducted assays with KRAS$^{G12D}$ and VHL or FRB constructs in the presence of the proteasomal inhibitor MG132 and the neddylation inhibitor MLN4924. The presence of both inhibitors substantially blocked rapamycin-mediated depletion of KRAS$^{G12D}$ by VHL-FRB, confirming that target depletion was indeed due to proteasomal degradation (*Figure 2H*). We conclude that the RiPA system is capable of inducing target ubiquitylation and predicting suitable target/E3 ligase pairs.

## RiPA can identify degradative E3 ligases not previously used for PROTACs

The RiPA system is well suited to quantify target protein degradation by VHL and Cereblon. However, these two E3 ligases are by far the most commonly used for PROTACs, so it is possible that they are exceptional in terms of potency or substrate promiscuity. We therefore wondered whether the RiPA system could detect and quantify target degradation by E3 ligases not currently used for PROTACs. We chose the E3 ligase FBXL12 as a candidate because its degradative function was recently suggested by pooled genome-wide screens (*Poirson et al., 2022*), but had not been exploited by PROTACs. We cloned FBXL12 into the FRB-containing entry vector (EV) and co-expressed it with WDR5 containing the lysine-free FKBP12 tag in HEK293 cells. Incubation with rapamycin-induced robust degradation of WDR5 as assessed by immunoblotting and luciferase activity assays (*Figure 3A and B*). Similar to our observation with VHL, the position of the luciferase in the WDR5-FKBP fusion protein is irrelevant to its activity and degradation by FBXL12 (*Figure 3—figure supplement 1A*). A direct comparison of WDR5 degradation induced by FBXL12 and VHL showed a strikingly superior activity of FBXL12 (*Figure 3C*). Conversely, FBXL12 exhibited comparable potency to VHL in targeting KRAS$^{G12D}$ (*Figure 2—figure supplement 1C* and *Figure 3—figure supplement 1B*).

Next, we wondered how the steric arrangement between a target protein and the FRB tag affects its degradation. So far, we have used a linker that is eight amino acids long (2x GSSG). Together with the size of FRB and FKBP12, this linker exceeds the length of most PROTACs (*Figure 3D*). We varied the linker length between WDR5 and FKBP12 from 0 to 8x GSSG and compared its effect on expression and degradation by FBXL12. Immunoblots showed that the cellular expression level of the WDR5 fusion protein declined with increasing linker length (*Figure 3E*). Strikingly, the construct without any linker between WDR5 and FKBP12 is not only the best expressed but also degraded most drastically by FBXL12 upon the addition of rapamycin (*Figure 3E and F*). We conclude that both tight and more flexible arrangements between the target protein and the dimerization tag allow quantification of E3 ligase activity, but that a seamless fusion shows the most efficient degradation, possibly mimicking the cooperativity of PROTAC-induced ternary complex formation.

Until now we have analyzed target degradation in the RiPA system by immunoblotting or by measuring luciferase activity after cell lysis. Since it is not only relevant how completely PROTACs induce target degradation, but also how fast, we tried to adapt the workflow for kinetic analyses. For this purpose, we replaced the luciferase substrate furimazine with endurazine, a substrate precursor that is continuously taken up and activated by the cells. We incubated WDR5 and FBXL12-expressing living cells with endurazine before adding rapamycin and measured luciferase activity every 15 min for 6 hr followed by every 30 min (*Figure 3G*). Rapamycin-mediated dimerization with FBXL12 induced a

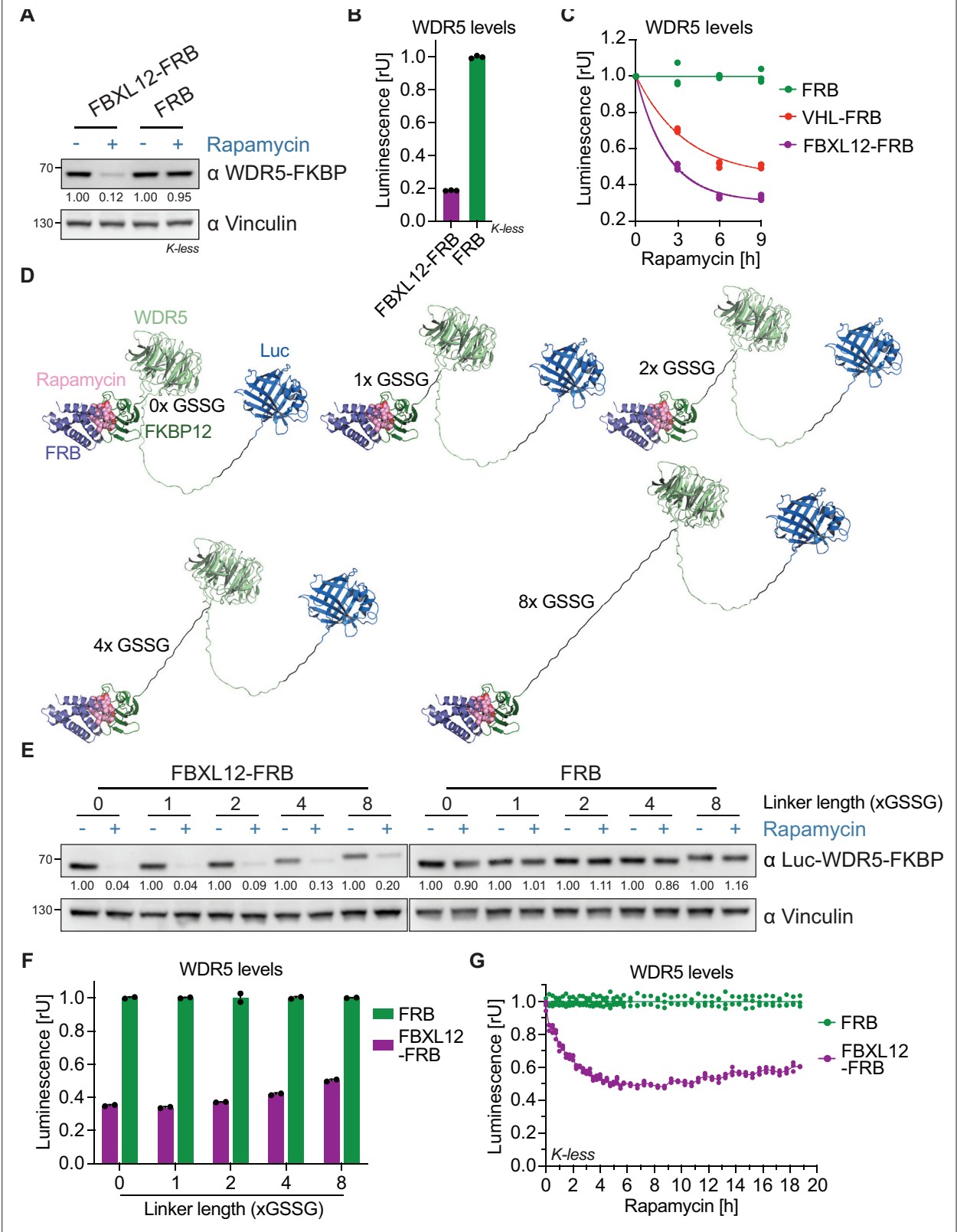

**Figure 3.** Rapamycin-induced proximity assay (RiPA) can identify degradative E3 ligases not previously used for proteolysis targeting chimeras (PROTACs). (**A**) Immunoblot of WDR5. HEK293 cells were transfected with WDR5-Luc-FKBP12(Kless) and FBXL12-FRB or FRB in a ratio of 1:10 and treated with 10 nM rapamycin for 8 hr. (**B**) WDR5 levels based on luciferase measurement. Luminescence of WDR5-Luc-FKBP12(Kless) in the same cells as in (**A**). Bars represent mean ± s.d. of n=3 replicates. (**C**) WDR5 levels based on luciferase measurement. HEK293 cells were transfected with WDR5-

*Figure 3 continued on next page*

*Figure 3 continued*

Luc-FKBP12(Kless) and FBXL12-FRB or VHL-FRB or FRB in a ratio of 1:1000 and treated with 10 nM rapamycin for the indicated time point. Data are shown as individual values (n=3 replicates), with a curve fitted to the mean across replicates. (**D**) Model of Luc-WDR5-FKBP12 constructs. Structure of Luc-WDR5-FKBP12 with indicated linkers between WDR5 and FKBP12 bound to rapamycin and FRB. The linker between Luc-WDR5 is always 2x GSSG. (**E**) Immunoblot of WDR5. HEK293 cells were transfected with Luc-WDR5-FKBP12 containing indicated linker length and FBXL12-FRB or FRB in the ratio of 1:100, expressed for ~24 hr, and treated with 10 nM rapamycin for 6 hr. (**F**) WDR5 levels based on luciferase measurement. Luminescence of Luc-WDR5-FKBP12 constructs as in cells from (**E**) and treated with 10 nM rapamycin for 8 hr. Bars represent mean ± s.d. of n=2 replicates. (**G**) WDR5 levels based on kinetic luciferase measurement. HEK293 cells were transfected with WDR5-Luc-FKBP12(Kless) and FBXL12-FRB or FRB in a ratio of 1:100, expressed for ~24 hr, treated with 10 nM rapamycin, and luminescence measured for 19 hr. Data are shown as individual values (n=3 replicates).

The online version of this article includes the following source data and figure supplement(s) for figure 3:

**Source data 1.** PDF file containing original western blot for *Figure 3A and E*, indicating the relevant bands.

**Source data 2.** Original files for western blot analysis displayed in *Figure 3A and E*.

**Figure supplement 1.** Rapamycin-induced proximity assay (RiPA) can identify degradative E3 ligases not previously used for proteolysis targeting chimeras (PROTACs).

robust and durable degradation of WDR5, reaching a maximum after 5 hr. We concluded that RiPA is able to estimate the degradation of 'novel' E3 ligases and allows kinetic measurements in living cells.

## Identification of degradation-inducing E3 ligases by designing a universal substrate

So far, our goal has been to optimize the RiPA system to identify suitable target/E3 ligase pairs by ensuring exclusive ubiquitylation of the specific protein target. A slightly different question is whether an E3 ligase can degrade a substrate at all, i.e., whether it can catalyze the addition of degradation-inducing ubiquitin chains. In fact, based on the Reactome Pathway and the UniProt database, only about 40% of 600 or more E3 ligases annotated in the human genome are believed to be associated with ubiquitin-proteasome system (*Schapira et al., 2019*). Therefore, we aimed to develop a universal substrate that is highly susceptible to ubiquitylation by increasing the available lysine receptor residues using two different strategies. First, we analyzed arginine residues on the surface of luciferase that can be mutated without a large impact on protein structure or substrate binding. Based on this consideration, we designed a mutant containing 5 (Luc^V1) lysine residues in addition to 7 surface residues of the wild-type protein (*Figures 2E and 4A*). As there were still surface patches without exposed lysines on Luc^V1, we carefully designed another mutant containing 6 additional lysine residues (Luc^V2), resulting in a total of 18 surface lysine residues on the protein (*Figure 4A*). Second, we fused luciferase to three different peptides containing a different number and arrangement of additional lysine residues (Luc^K3, Luc^K6, or Luc^K12; *Figure 4B*).

All luciferase mutants except Luc^K12, which contains 12 consecutive lysine residues in the peptide tail, are expressed to a level that allows robust measurements (*Figure 4C*). We then co-expressed these mutants with FBXL12 and observed that the luciferase versions with additional lysine residues on the cell surface (Luc^V1, Luc^V2) were degraded at the same rate and to a similar extent as wild-type luciferase. In contrast, the additional lysines in Luc^K3 and Luc^K6 dramatically increased their degradation by FBXL12 (*Figure 4D*). Kinetic measurements in living cells revealed that more than 60% of Luc^K6 is already degraded 2 hr after the addition of rapamycin (*Figure 4E*). Taken together, these results demonstrate that mutant versions of luciferase can be used as universal reporters to stratify E3 ligases in terms of their degradative potential.

## Discussion

While PROTACs have revolutionized drug development, the design of such molecules, including the choice of appropriate E3 ligases, is still largely empirical (*Bond and Crews, 2021*). A growing body of literature indicates that often a specific E3 ligase cannot be used for PROTAC-induced degradation of a given target protein and that certain target/E3 ligase pairs are more suitable than others (*Donovan et al., 2020*; *Zhang et al., 2023*; *Yang et al., 2023*; *Steinebach et al., 2020*). Therefore, we developed a RiPA to quantitatively measure the degradation of specific targets by E3 ligases when brought together. We have carefully designed, tested, and modified the RiPA system to demonstrate that it has features that allow it to be used for E3 ligase selection in PROTAC development campaigns.

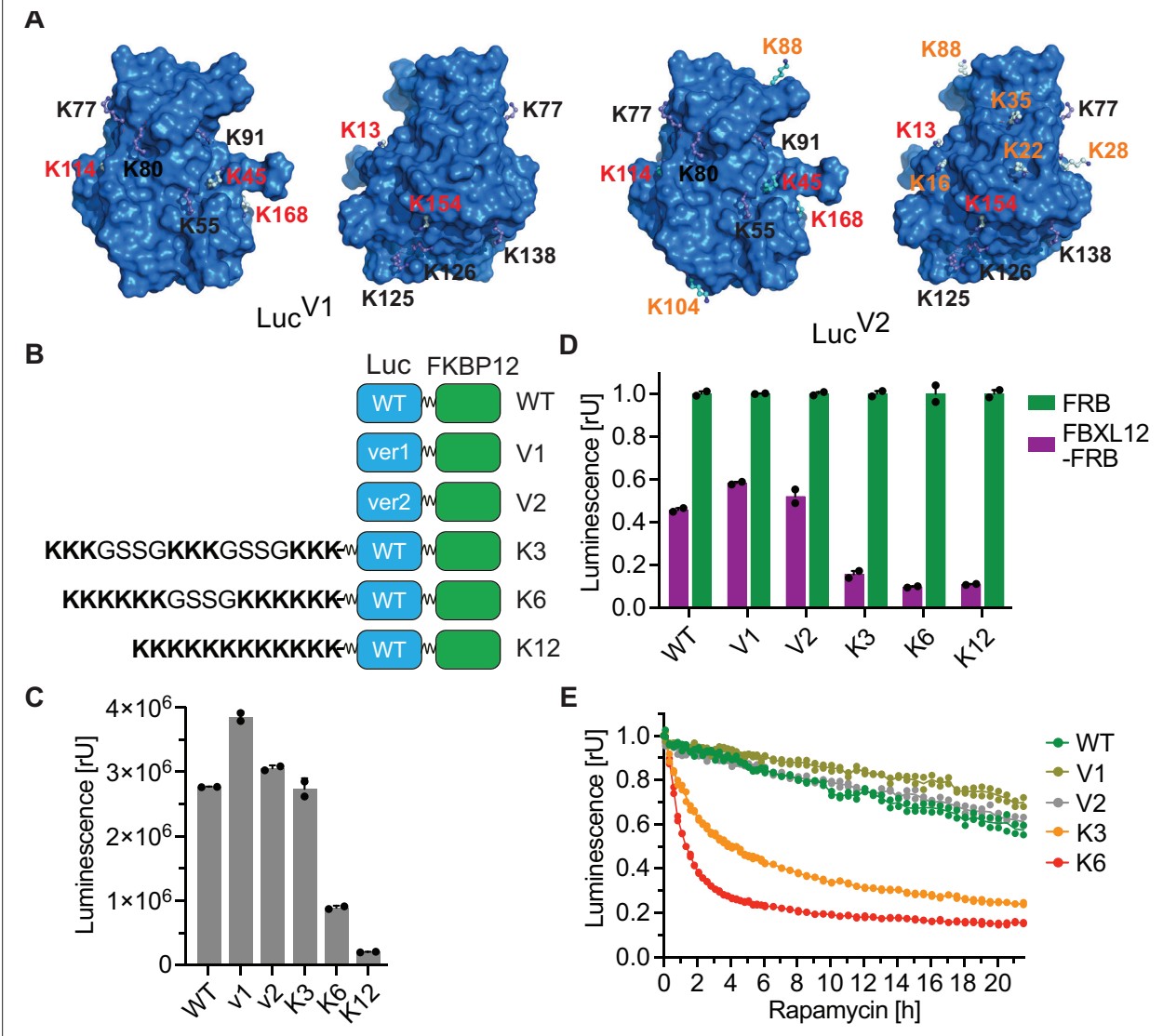

**Figure 4.** Identification of degradation-inducing E3 ligases by designing a universal substrate. (**A**) Model of lysine-rich luciferase. Structure of mutated luciferase with 5 additional (Luc[V1]) and 12 additional (Luc[V2]) lysines as compared to 7 lysine residues of wild-type luciferase. The lysine residues from WT (black), Luc[V1] (red), and Luc [V2] (orange; red as in V1) are labeled. (**B**) Scheme of wild-type luciferase (WT) or lysine-rich luciferase (V1, V2, K3, K6, and K12) containing constructs. (**C**) Luciferase measurement. HEK293 cells were co-transfected with Luc-FKBP12 constructs as shown in (**B**) and FRB, expressed for ~24 hr, and luminescence was compared. Bars represent mean ± s.d. of n=2 replicates. (**D**) Luciferase measurement. HEK293 cells were transfected with the indicated versions of Luc-FKBP12 and FBXL12-FRB or FRB in a ratio of 1:100, expressed for ~24 hr, and treated with 10 nM rapamycin for 8 hr. Bars represent mean ± s.d. of n=2 replicates. (**E**) Kinetic luminescence measurement. HEK293 cells expressing constructs as described in (**D**) were treated with 10 nM rapamycin or vehicle and luminescence was monitored for 22 hr. Data are shown as individual values (n=2 replicates).

The online version of this article includes the following figure supplement(s) for figure 4:

**Figure supplement 1.** Timeline for rapamycin-induced proximity assay (RiPA).

First, we were able to show that RiPA is in principle able to induce target degradation by analyzing the degradation of WDR5 when brought into proximity with VHL. Second, we constructed a set of lentiviral vectors that can be used for straightforward cloning of the E3 ligase and target candidates, and can be delivered to eukaryotic cells by both viral transduction and transient transfection. While viral transduction allows the use of difficult-to-transfect cell lines and the careful characterization of a few specific target/E3 ligase pairs, transient transfection allows broad screening, potentially with all of more than 600 annotated E3 ligases under S1 conditions. We observed that the plasmid encoding the candidate E3 ligase must be transfected in at least 10-fold excess over the target

protein-containing plasmid for efficient degradation. Third, we observed that both, FRB and the minimal luciferase used for dimerization and detection of the target protein respectively, still function when all lysines are mutated to arginine. This ensures that RiPA only reports degradation of the target protein when it is directly ubiquitylated since ubiquitylation of the reporter can be excluded. We envision that this setting will be valuable for identifying the most suitable E3 ligase candidates for PROTACs aimed at specific proteins, and for guiding E3 ligase selection when screening for molecular glues targeting specific E3 ligase and protein pairs. Conversely, we have also constructed reporters that contain lysines in addition to their endogenous lysines and have observed drastically increased degradation in some cases. These tools can be used to analyze whether an uncharacterized E3 ligase can in principle add degradation-inducing ubiquitin chains, thus complementing data from pooled genetic screens with fluorescent reporters (*Zeng et al., 2020*). Comparing degradation with different types of these universal reporters allows estimation of the degree of E3 ligase promiscuity that is desired for PROTACs and thus identification of 'PROTACable' E3 ligases. Finally, by combining luminescent reporters with different substrates, RiPA can be used to precisely quantify target protein degradation in living cells and in extracts of lysed cells. Taken together, RiPA is a versatile system that is easy to use with standard molecular biology laboratory equipment. The timeline and hands-on time required for RiPA with five or less targets and E3 ligases are illustrated in *Figure 4—figure supplement 1A*.

While this technical study focused on the RiPA system as a resource, we made some notable observations. The most striking finding was the lack of degradation of WDR5 by Cereblon. While VHL efficiently reduced WDR5 levels by more than 50% within 6 hr of rapamycin addition, Cereblon did not induce WDR5 degradation under any condition or time frame. The lack of WDR5 degradation is not due to a general inability of Cereblon to induce protein degradation RiPA conditions, since Cereblon efficiently decreased Aurora A levels. This is noteworthy because it recapitulates our experience and that of others with the WDR5 and Aurora A PROTACs: while Cereblon-based PROTACs were able to efficiently degrade Aurora A (*Donovan et al., 2020*; *Adhikari et al., 2020*; *Wang et al., 2021*; *Rishfi et al., 2023*; *Liu et al., 2022*), they were either nonfunctional (*Dölle et al., 2021*) or much less potent (*Yu et al., 2023*; *Yu et al., 2021*) at degrading WDR5 compared to VHL-based PROTACs. A possible reason for the selective potency of both E3 ligases to degrade WDR5 by PROTACs could be their different ability to bind to WDR5 through protein-protein interactions and thus support ternary complex formation. Indeed, WDR5 could be co-crystallized with a variety of PROTACs and the respective E3 ligase, resulting in ternary complexes with a range of interface sizes between the two proteins. The most potent PROTACs, however, induced the largest interaction area, underlining the importance of achieving extensive protein-protein interactions for the most efficient target degradation (*Schwalm et al., 2023*). Strikingly, WDR5 degradation also decreased in the RiPA setting as the linkers between WDR5 and FKBP12 became longer, with the WDR5-FKBP12 construct without any linker being the most efficient. We conclude that RiPA closely recapitulates the E3 ligase selectivity of WDR5 PROTACs and hypothesize that it can, in principle, predict E3 ligases capable of forming ternary complexes with specific target proteins.

Another finding of our study is that the SCF (SKP1-CUL1 F-box protein)-type FBXL12 can degrade artificial substrates in the context of the RiPA system. We consider this to be relevant since the vast majority of PROTACs today use one of the two E3 ligases Cereblon or VHL. Therefore, it was possible that these two E3 ligases are particularly or even exclusively suited for the development of PROTACs, e.g., due to their high potency or promiscuity. Since FBXL12-mediated degradation of WDR5 is even more efficient than degradation by Cereblon, and since FBXL12 can induce degradation of every target protein tested in our study, our data suggest that FBXL12 may be a promising new E3 ligase for PROTAC development.

## Limitations of the study

While we consider RiPA to be very helpful in guiding PROTAC development campaigns, its results will not be fully generalizable to the characteristics of PROTACs, as it is conceivable for this system to make false-negative and false-positive predictions. Thus, suitable target/E3 ligase pairs can fail in the RiPA setting because protein tags interfere with their interaction or reduce the activity of the E3 ligases. While our system offers easy testing of different tagging approaches and due to its simple workflow facilitates the rapid characterization of novel E3 ligases across multiple targets, it is currently

not optimized for high-throughput evaluation of all 600+ E3 ligases. Achieving such scale would necessitate further adaptations, including the incorporation of pooled experimental strategies.

Conversely, it is also conceivable that an E3 ligase that can efficiently decrease the levels of a particular target in the RiPA setting may be less suitable for PROTACs, since PROTACs that mimic the steric interaction of the target/E3 pair may not be easily identified in the chemical space. However, the RiPA system can certainly identify inappropriate E3 ligases that are not in the same cellular compartment as the target or that cannot add a degradation-inducing ubiquitin chain. Since RiPA also correctly predicted the differential suitability of VHL and Cereblon to degrade WDR5, we anticipate that it will be very helpful in streamlining chemical efforts in PROTAC development campaigns, both for PROTACs based on established and novel E3 ligases.

# Materials and methods

## Key resources table

| Reagent type (species) or resource | Designation | Source or reference | Identifiers | Additional information |
|---|---|---|---|---|
| Antibody | anti-WDR5 (G-9) (mouse monoclonal) | Santa Cruz Biotechnology | RRID: AB_3331659 | WB (1:1000) |
| Antibody | anti-AuroraA/AIK (rabbit, polyclonal) | Cell Signaling Technology | RRID: AB_2061342 | WB (1:1000) |
| Antibody | anti-VHL (VHL40) (mouse monoclonal) | Santa Cruz Biotechnology | RRID: AB_2215955 | WB (1:1000) |
| Antibody | anti-CRBN (D8H3S) (rabbit, monoclonal) | Cell Signaling Technology | RRID: AB_2799810 | WB (1:1000) |
| Antibody | anti-vinculin (mouse monoclonal) | Sigma-Aldrich | RRID: AB_477629 | WB (1:2000) |
| Antibody | ECL-Anti-rabbit IgG Horseradish Peroxidase | GE Healthcare | Cat# NA934V | WB (1:7500) |
| Antibody | ECL-Anti-mouse IgG Horseradish Peroxidase | GE Healthcare | Cat# NA931V | WB (1:7500) |
| Chemical compound, drug | Rapamycin | Selleckchem | Cat# S1039 | |
| Chemical compound, drug | MG132 | Calbiochem/Merck | Cat# 474790 | |
| Chemical compound, drug | Pevonedistat (MLN4924) | Selleckchem | Cat# S7109 | |
| Chemical compound, drug | Protease Inhibitor Cocktail | Sigma-Aldrich | Cat# P8340 | |
| Chemical compound, drug | Phosphatase Inhibitor Cocktail 2 | Sigma-Aldrich | Cat# P5726 | |
| Chemical compound, drug | Phosphatase Inhibitor Cocktail 3 | Sigma-Aldrich | Cat# P0044 | |
| Peptide, recombinant protein | Phusion High-Fidelity DNA Polymerase | Thermo Fisher Scientific | Cat# F530L | |
| Chemical compound, drug | Immobilon Western Chemiluminescent HRP Substrate | Merck Millipore | Cat#WBKLS0500 | |
| Peptide, recombinant protein | NanoGlo Endurazine Live Cell Substrate | Promega | Cat# N2571 | |
| Peptide, recombinant protein | Phusion Plus DNA Polymerase | Thermo Fisher Scientific | Cat# F630L | |
| Peptide, recombinant protein | AgeI-HF | New England BioLabs | Cat# R3552L | |
| Peptide, recombinant protein | AscI | New England BioLabs | Cat# R0558L | |

*Continued on next page*

*Continued*

| Reagent type (species) or resource | Designation | Source or reference | Identifiers | Additional information |
|---|---|---|---|---|
| Peptide, recombinant protein | BamHI-HF | New England BioLabs | Cat# R3136L | |
| Peptide, recombinant protein | EcoRI-HF | New England BioLabs | Cat# R3101L | |
| Peptide, recombinant protein | MluI-HF | New England BioLabs | Cat# R3198L | |
| Peptide, recombinant protein | SpeI-HF | New England BioLabs | Cat# R3133L | |
| Peptide, recombinant protein | XhoI | New England BioLabs | Cat# R0146L | |
| Commercial assay or kit | Nano-Glo Luciferase Assay | Promega | Cat# N1120 | |
| Commercial assay or kit | GeneJET Gel Extraction Kit | Thermo Fisher Scientific | Cat# K0692 | |
| Commercial assay or kit | PureLink HiPure Plasmid Maxiprep Kit | Invitrogen | Cat# K210007 | |
| Cell line (Homo-sapiens) | human embryonic kidney 293T | ATCC | CRL-3216 | |
| Recombinant DNA reagent | pRRL_puro (plasmid) | PMID:25043018 | | |
| Recombinant DNA reagent | pRRL_hygro (plasmid) | PMID:25043018 | | |
| Recombinant DNA reagent | pRRL_puro_WDR5-Luc-FKBP12 (plasmid) | This paper | | See Materials and methods, Section 'Cloning of Target-FKBP12 constructs' |
| Recombinant DNA reagent | pRRL_puro_Luc-WDR5-FKBP12 (plasmid) | This paper | | See Materials and methods, Section 'Cloning of Target-FKBP12 constructs' |
| Recombinant DNA reagent | pRRL_hygro_FRB-VHL (plasmid) | This paper | | See Materials and methods, Section 'Cloning of E3-FRB constructs' |
| Recombinant DNA reagent | pRRL_hygro_VHL-FRB (plasmid) | This paper | | See Materials and methods, Section 'Cloning of E3-FRB constructs' |
| Recombinant DNA reagent | pRRL_hygro_FRB (plasmid) | This paper | | See Materials and methods, Section 'Cloning of E3-FRB constructs' |
| Recombinant DNA reagent | pRRL_hygro_CRBN-FRB (plasmid) | This paper | | See Materials and methods, Section 'Cloning of E3-FRB constructs' |
| Recombinant DNA reagent | pRRL_puro_AURKA-Luc-FKBP12 (plasmid) | This paper | | See Materials and methods, Section 'Cloning of Target-FKBP12 constructs' |
| Recombinant DNA reagent | pRRL_puro_WDR5-Luc-FKBP12(Kless) (plasmid) | This paper | | See Materials and methods, Section 'Cloning of Target-FKBP12 constructs' |
| Recombinant DNA reagent | pRRL_puro_AURKB-Luc-FKBP12(Kless) (plasmid) | This paper | | See Materials and methods, Section 'Cloning of Target-FKBP12 constructs' |
| Recombinant DNA reagent | pRRL_puro_KRAS$^{G12D}$-Luc-FKBP12(Kless) (plasmid) | This paper | | See Materials and methods, Section 'Cloning of Target-FKBP12 constructs' |

*Continued on next page*

*Continued*

| Reagent type (species) or resource | Designation | Source or reference | Identifiers | Additional information |
|---|---|---|---|---|
| Recombinant DNA reagent | pRRL_hygro_FBXL12-FRB (plasmid) | This paper | | See Materials and methods, Section 'Cloning of E3-FRB constructs' |
| Recombinant DNA reagent | pRRL_puro_C-term_Luc-FKBP12_EV (plasmid) | This paper | | See Materials and methods, Section 'Cloning of Target-FKBP12 constructs' |
| Recombinant DNA reagent | pRRL_puro_C-term_Luc-FKBP12(Kless)_EV (plasmid) | This paper | | See Materials and methods, Section 'Cloning of Target-FKBP12 constructs' |
| Recombinant DNA reagent | pRRL_hygro_E3-FRB_EV (plasmid) | This paper | | See Materials and methods, Section 'Cloning of E3-FRB constructs' |
| Recombinant DNA reagent | pRRL_hygro_FRB-E3_EV (plasmid) | This paper | | See Materials and methods, Section 'Cloning of E3-FRB constructs' |
| Recombinant DNA reagent | pRRL_puro_Luc-WDR5-FKBP12_0xGSSG (plasmid) | This paper | | See Materials and methods, Section 'Cloning of Target-FKBP12 constructs' |
| Recombinant DNA reagent | pRRL_puro_Luc-WDR5-FKBP12_1xGSSG (plasmid) | This paper | | See Materials and methods, Section 'Cloning of Target-FKBP12 constructs' |
| Recombinant DNA reagent | pRRL_puro_Luc-WDR5-FKBP12_2xGSSG (plasmid) | This paper | | See Materials and methods, Section 'Cloning of Target-FKBP12 constructs' |
| Recombinant DNA reagent | pRRL_puro_Luc-WDR5-FKBP12_4xGSSG (plasmid) | This paper | | See Materials and methods, Section 'Cloning of Target-FKBP12 constructs' |
| Recombinant DNA reagent | pRRL_puro_Luc-WDR5-FKBP12_8xGSSG (plasmid) | This paper | | See Materials and methods, Section 'Cloning of Target-FKBP12 constructs' |
| Recombinant DNA reagent | pRRL_puro_Luc-FKBP12 (plasmid) | This paper | | See Materials and methods, Section 'Cloning of Luc-FKBP12 universal substrate constructs' |
| Recombinant DNA reagent | pRRL_puro_Luc_V1-FKBP12 (plasmid) | This paper | | See Materials and methods, Section 'Cloning of Luc-FKBP12 universal substrate constructs' |
| Recombinant DNA reagent | pRRL_puro_Luc_V2-FKBP12 (plasmid) | This paper | | See Materials and methods, Section 'Cloning of Luc-FKBP12 universal substrate constructs' |
| Recombinant DNA reagent | pRRL_puro_Luc_K3-FKBP12 (plasmid) | This paper | | See Materials and methods, Section 'Cloning of Luc-FKBP12 universal substrate constructs' |
| Recombinant DNA reagent | pRRL_puro_Luc_K6-FKBP12 (plasmid) | This paper | | See Materials and methods, Section 'Cloning of Luc-FKBP12 universal substrate constructs' |
| Recombinant DNA reagent | pRRL_puro_Luc_K12-FKBP12 (plasmid) | This paper | | See Materials and methods, Section 'Cloning of Luc-FKBP12 universal substrate constructs' |
| Sequence-based reagent | Oligonucleotides used for PCR and cloning | Sigma-Aldrich | | See **Supplementary file 1** |
| Sequence-based reagent | IDT G-blocks (double stranded linear DNA) | Integrated DNA technologies | | See **Supplementary file 2** |

*Continued on next page*

*Continued*

| Reagent type (species) or resource | Designation | Source or reference | Identifiers | Additional information |
|---|---|---|---|---|
| Software, algorithm | GraphPad PRISM 10.1.1 | GraphPad | https://www.graphpad.com/ | |
| Software, algorithm | ImageJ 1.52 | PMID:22930834 | https://imagej.nih.gov/ij/ | |
| Software, algorithm | PyMOL Molecular Graphics System | Schrödinger, LLC | https://pymol.org/ | |

## Cell line and cell culture

Human HEK293T (female, embryo kidney) cells were cultured in DMEM medium (Thermo Fisher Scientific) supplemented with 10% FBS (Capricorn Scientific) and 1% penicillin/streptomycin (Sigma). Cells were grown at 37°C under 5% $CO_2$. The cells were routinely checked for mycoplasma contamination in a PCR-based assay and found negative.

## Cloning of target-FKBP12 constructs

WDR5-Luc-FKBP12, AURKA-Luc-FKBP12, and Luc-WDR5-FKBP12 vectors were cloned by PCR amplification of the template vector for each fragment, WDR5 or AURKA, Luc, and FKBP12. A flexible linker (2x GSSG) between target-Luc and Luc-FKBP12 or Luc-target and target-FKBP12 was incorporated in the primers (Sigma) used. The corresponding fragments were digested with appropriate restriction enzymes and ligated into pRRL_puro. For the WDR5-Luc-FKBP12(Kless) construct, first, a gBlock (IDT) containing MCS, the mutated Luc (K55R, K77R, K80R, K91R, K125R, K126R and K138R), a linker (2x GSSG), and the mutated FKBP12 (K18R, K35R, K36R, K45R, K48R, K53R, K74R, and K106R) was ordered. AscI/BamHI was then used to insert the gBlock into pRRL_puro, resulting in the C-term_Luc-FKBP12-Kless EV. Subsequently, the WDR5 fragment was PCR-amplified and inserted into the EV using AgeI/SpeI restriction sites to get WDR5-Luc-FKBP12(Kless). The KRASG12D-Luc-FKBP12(Kless) vector was constructed by digesting a gBlock (IDT) and ligating it into the C-term_Luc-FKBP12-Kless EV using AgeI/SpeI restriction sites. Similarly, the C-term_Luc-FKBP12 EV was produced by PCR amplification of Luc and FKBP12 fragments, respectively, followed by an overlapping PCR to generate a Luc-FKBP12 insert. The C-term_Luc-FKBP12 EV was then obtained by replacing Luc-FKBP12(Kless) of the C-term_Luc-FKBP12-Kless EV.

For cloning of the Luc-WDR5-FKBP12 constructs with different GSSG linker lengths, the Luc-WDR5 fragment was PCR-amplified using pRRL_puro_Luc-WDR5-FKBP12 as a template. The reverse primer binding to WDR5 contained the required linkers (0x or 1x or 2x or 4x or 8x GSSG). The PCR product containing various linkers was then inserted into pRRL_puro_Luc-WDR5-FKBP12 plasmid backbone after removing Luc-WDR5 using AgeI/MluI restriction sites.

## Cloning of Luc-FKBP12 universal substrate constructs

The Luc-FKBP12 vector was cloned by PCR amplification of Luc and FKBP12 fragments and subsequent ligation into the pRRL_puro backbone. To clone the lysine-rich Luc mutants, Luc[V1] and Luc[V2], Luc_ver1 (R13K, R45K, R114K, R154K, R168K) and Luc_ver2 (R13K, A16K, Q22K, G28K, N35K, R45K, H88K, V104K, R114K, R154K, R168K) were ordered as gBlocks. Luc was replaced in pRRL_puro_Luc-FKBP12 with Luc_ver1 and Luc_ver2 using AgeI/MluI restriction sites. For Luc[K3], Luc[K6], and Luc[K12] cloning, two oligos (top and bottom) were designed that contain the additional lysine residues and overhangs for AgeI/SpeI restriction site. The oligos were hybridized and ligated into the pRRL_puro_C-term_Luc-FKBP12 EV using AgeI/SpeI sites.

## Cloning of E3-FRB constructs

VHL-FRB, FRB-VHL, CRBN-FRB, FBXL12-FRB, and FRB vectors were cloned by PCR amplification of VHL, CRBN, FBXL12, or FRB fragments from template plasmids or cDNA (FBXL12). A flexible linker (2x GSSG) between the respective E3 ligase and FRB was incorporated with the primers used. The corresponding fragments were digested with appropriate restriction enzymes and ligated into pRRL_hygro backbone using AgeI/SpeI sites. For E3-FRB EV and FRB-E3 EV, gBlocks containing FRB, a linker (2x GSSG), and an MCS were ordered and inserted into pRRL_hygro using AscI/BamHI sites.

## Rapamycin-induced proximity assays

HEK293T cells were plated at a density of $6.5 \times 10^5$ in 2 ml media per well in a six-well plate. The cells were allowed to attach and recover for at least 6 hr before transfection. For transfection, two Eppendorf tubes with 140 µl OptiMEM (Thermo Fisher Scientific) each were prepared with either 6 µl polyethylenimine (PEI; Sigma) or the appropriate amounts of FKBP12/FRB plasmid pairs. A total of about 1.6 µg of plasmids in the required ratio (FKBP12:FRB) was used per transfection. The contents were mixed well, centrifuged, and incubated for 5 min at room temperature (RT) before the plasmid mixture was added to the PEI mixture. After 20 min of incubation at RT, the plasmid-PEI mixture was added to the attached cells. The cells were left in the incubator for at least 20 hr for protein expression. Next, the cells were trypsinized, collected in media, centrifuged, and resuspended in 3 ml of media (DMEM for endpoint; assay medium, OptiMEM+4% FBS+1% P/S+15 mM HEPES for kinetic). For endpoint and kinetic luciferase measurement, 45 µl of the cell suspension was seeded into two sets of replicates (for control and rapamycin) per transfection condition in a black 96-well plate. For endpoint western blot (WB), the rest of the cell suspension was equally divided and seeded into two wells of a six-well plate per transfection condition while maintaining a final volume of 2 ml with the media. The plates were left in incubator overnight. The next day, rapamycin or control (DMSO) treatment for endpoint measurements was performed by adding either rapamycin or a corresponding amount of DMSO to a final concentration of 10 or 100 nM. For the luciferase measurement rapamycin was prediluted in assay media. After the treatment time, the cells were lysed with the Nano-Glo Luciferase Assay Reagent (Promega) according to the manufacturer's protocol, and luminescence was measured using the Tecan Spark Multiplate reader (Tecan) with an integration time of 1 s. Similarly, for WB the cells were lysed in RIPA lysis buffer (50 mM HEPES pH 7.9, 140 mM NaCl, 1 mM EDTA, 1% Triton X-100, 0.1% SDS, 0.1% sodium deoxycholate) containing protease and phosphatase inhibitor (Sigma).

For the kinetic measurement, 2–3 hr prior to rapamycin addition, 15 µl of 5x Endurazine (prediluted in assay medium) was added. After 2–3 hr of incubation, 15 µl of 50 nM rapamycin (prediluted in assay medium) was added to the corresponding wells. Finally, kinetic measurement with 15–45 min intervals was initiated on the Tecan Spark Multiplate reader at 37°C/5% $CO_2$ in the presence of a humidity cassette for 20 hr.

## Western blot

Cells treated with DMSO or rapamycin were lysed in RIPA lysis buffer for 30 min at 4°C in rotator. The cell debris was cleared from lysate by centrifugation and BCA assay was done to measure protein concentration. Equal amounts of protein per sample were separated by Bis-TRIS PAGE and transferred to PVDF membranes (Millipore). The membranes were blocked with blocking solution (5% milk in TBS-T; 20 mM Tris HCl, pH 7.5, 150 mM NaCl, and 0.1% Tween 20), cut into pieces for different proteins and incubated with corresponding primary antibodies at 4°C overnight. Next, the membranes were washed with TBS-T and incubated with HRP-labeled secondary antibodies at RT for 1 hr. Visualization was done with chemiluminescent HRP substrate (Millipore) in LAS4000 Mini (Fuji).

## Protein structures

For modeling and visualization all crystal structures indicated below were prepared using the Molecular Operating Environment (MOE) (*ChemicalComputingGroup ULC, 2022*).

For the minimal luciferase, PDB structure 7SNS (resolution of 1.55 Å) was used. Only chain A was prepared by modeling missing side chains, adding hydrogen atoms at pH 7 with Protonate 3D (*Labute, 2009*) and renumbering the residues starting at 1 with the first resolved residue (methionine). This residue was then removed as it was not present in the cloned constructs. Buffer and water molecules were also removed from the structure.

The structure of FKBP12, FRB, and rapamycin was generated by combining protein residues from structure 3FAP (*Liang et al., 1999*) (1.85 Å) and rapamycin from structure 1NSG (*Liang et al., 1999*) (2.20 Å). The higher occupancy conformation of R175 in chain B was retained, and the system was protonated at pH 7. To match the cloned constructs, the two N-terminal residues (V and A) were removed, and a C-terminal lysine was added. Water molecules were deleted from both structures.

Since no full-length structure of WDR5 was available, residues 32–334 from a crystal structure (PDB: 2H14 [*Couture et al., 2006*], resolution 1.48 Å) were combined with residues 1–31 of the prediction AF-P61964-F1 from the AlphaFold2 structure database (*Jumper et al., 2021*; *Varadi et al., 2022*).

The predicted residues were modeled with low to very low confidence but were considered suitable to indicate the approximate distance between WDR5 and minimal luciferase when linked together. All water molecules were removed from the system.

The proteins were locally minimized to a gradient of 0.1 kcal/mol*Å using the AMBER14:EHT force field and tether restraints on all atoms (σ: 0.5 Å). In order to estimate the distances between the individual proteins in the cloned constructs, GSSG linkers were modeled and connected to the respective systems. The linkers were generated in a linear fashion to indicate the maximum possible distance between the proteins and the relative positions of attachment points. However, they do not make assumptions about possible or stable conformations. To introduce more lysine residues to the minimal luciferase, potential mutation sites were selected by visually inspecting the prepared structure.

All figures of protein structures were rendered using PyMOL 2.5.7 (*Schrödinger, 2000*).

## Resource availability

Plasmids generated in this study are available on request from the lead contact. Further information and requests for resources and reagents should be directed to the lead contact, Elmar Wolf (elmar.wolf@biochem.uni-kiel.de).

## Acknowledgements

We thank André Kutschke for excellent technical support. This work was supported by grants from the German Research Foundation (DFG, WO 2108/2-1, and CRC 387 to EW and GRK 2243 to BA), the European Research Council (ERC, PROTAC-PDAC to EW), the German Cancer Aid (funding of TACTIC to EW), and the Federal Ministry of Education and Research (BMBF, DEGRON to EW).

## Additional information

### Competing interests

Bikash Adhikari, Elmar Wolf: The University of Würzburg has filed a patent (EP4464788) for the RiPA system described in the study and EW and BA are listed as inventors. The other authors declare that no competing interests exist.

### Funding

| Funder | Grant reference number | Author |
| --- | --- | --- |
| Deutsche Forschungsgemeinschaft | WO 2108/2-1 | Elmar Wolf |
| Deutsche Forschungsgemeinschaft | CRC 387 | Elmar Wolf |
| European Research Council | PROTAC-PDAC | Elmar Wolf |
| Deutsche Krebshilfe | TACTIC | Elmar Wolf |
| Bundesministerium für Bildung und Forschung | DEGRON | Elmar Wolf |
| German Research Foundation | GRK 2243 | Bikash Adhikari |

The funders had no role in study design, data collection and interpretation, or the decision to submit the work for publication.

### Author contributions

Bikash Adhikari, Conceptualization, Formal analysis, Funding acquisition, Validation, Investigation, Methodology, Writing – original draft, Writing – review and editing; Katharina Schneider, Investigation, Methodology, Writing – original draft, Writing – review and editing; Mathias Diebold, Investigation, Methodology, Writing – original draft; Christoph Sotriffer, Supervision, Writing – original draft;

Elmar Wolf, Conceptualization, Supervision, Funding acquisition, Writing – original draft, Writing – review and editing

### Author ORCIDs
Katharina Schneider http://orcid.org/0009-0009-4594-1609
Elmar Wolf https://orcid.org/0000-0002-5299-6335

Reviewer #1 (Public review): https://doi.org/10.7554/eLife.98450.3.sa1
Reviewer #2 (Public review): https://doi.org/10.7554/eLife.98450.3.sa2
Author response https://doi.org/10.7554/eLife.98450.3.sa3

## Additional files

### Supplementary files
• Supplementary file 1. Amino acid sequences. The table contains all the amino acid sequences of the constructs used in this study.

• Supplementary file 2. Oligonucleotide sequences. The table contains all oligonucleotide sequences, primers, and gBlocks used in this study.

• MDAR checklist

### Data availability
All data generated or analysed during this study are included in the manuscript and supporting files.

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
