## [Editor Report · eLife Assessment]

The study describes a **valuable** new technology in the field of targeted protein degradation that allows identification of E3-ubiquitin ligases that target a protein of interest. The presented data are **convincing**, however, additional work will be needed to optimize for high-throughput evaluation. This technology will therefore serve the community in the initial stages of developing targeted protein degraders.

---

## [Referee Report · Reviewer #1 (Public review)]

Summary:

PROTACs are heterobifunctional molecules that utilize the Ubiquitin Proteasome System to selectively degrade target proteins within cells. Upon introduction to the cells, PROTACs capture the activity of the E3 ubiquitin ligases for ubiquitination of the targeted protein, leading to its subsequent degradation by the proteasome. The main benefit of PROTAC technology is that it expands the "druggable proteome" and provides numerous possibilities for therapeutic use. However, there are also some difficulties, including the one addressed in this manuscript: identifying suitable target-E3 ligase pairs for successful degradation. Currently, only a few out of about 600 E3 ligases are used to develop PROTAC compounds, which creates the need to identify other E3 ligases that could be used in PROTAC synthesis. Testing the efficacy of PROTAC compounds has been limited to empirical tests, leading to lengthy and often failure-prone processes. This manuscript addressed the need for faster and more reliable assays to identify the compatible pairs of E3 ligases-target proteins. The authors propose using the RiPA assay, which depends on rapamycin-induced dimerization of FKBP12 protein with FRB domain. The PROTAC technology is advancing rapidly, making this manuscript both timely and essential. The RiPA assay might be useful in identifying novel E3 ligases that could be utilized in PROTAC technology. Additionally, it could be used at the initial stages of PROTAC development, looking for the best E3 ligase for the specific target.

The authors described an elegant assay that is scalable, easy-to-use and applicable to a wide range of cellular models. This method allows for the quantitative validation of the degradation efficacy of a given pair of E3 ligase-target protein, using luciferase activity as a measure. Importantly, the assay also enables the measurement of kinetics in living cells, enhancing its practicality.

Strengths:

(1) The authors have addressed the crucial needs that arise during PROTAC development. In the introduction, they nicely describe the advantages and disadvantages of the PROTAC technology and explain why such an assay is needed.

(2) The study includes essential controls in experiments (important for generating new assay), such as using the FRB vector without E3 ligase as a negative control, testing different linkers (which may influence the efficacy of the degradation), and creating and testing K-less vectors to exclude the possibility of luciferase or FKBP12 ubiquitination instead of WDR5 (the target protein). Additionally, the position of the luc in the FKBP12 vector and the position of VHL in the FRB vector are tested. Different E3 ligases are tested using previously identified target proteins, confirming the assay's utility and accuracy.

(3) The study identified a "new" E3 ligase that is suitable for PROTAC technology (FBXL).

Weaknesses:

It is not clear how feasible it would be to adapt the assay for high-throughput screens.

Comments on revisions:

The authors have addressed my previous concerns and made changes to the manuscript, resulting in a well-written paper.

---

## [Referee Report · Reviewer #2 (Public review)]

Summary:

Adhikari and colleagues developed a new technique, rapamycin-induced proximity assay (RiPA), to identify E3-ubiquitin (ub) ligases of a protein target, aiming at identifying additional E3 ligases that could be targeted for PROTAC generation or ligases that may degrade a protein target. The study is timely, as expanding the landscape of E3-ub ligases for developing targeted degraders is a primary direction in the field.

Strengths:

(1) The study's strength lies in its practical application of the FRB:FKBP12 system. This system is used to identify E3-ub ligases that would degrade a target of interest, as evidenced by the reduction in luminescence upon the addition of rapamycin. This approach effectively mimics the potential action of a PROTAC.

Weaknesses:

(1) While the technique shows promise, its application in a discovery setting, particularly for high-throughput or unbiased E3-ub ligase identification, may pose challenges. The authors now discuss these potential difficulties providing a more comprehensive understanding of RiPA's limitations.

(2) While RiPA will help identify E3 ligases, PROTAC design would still be empirical. The authors provide some discussion of this limitation.

Comments on revisions:

I thank the authors for addressing my prior concerns. I would recommend that individual replicate values are plotted in all the mean -/+ s.d or sem graphs.

---

## [Author Response]

The following is the authors’ response to the original reviews.

First of all, we would like to thank the reviewers for their very constructive comments, which helped us to improve the manuscript! In response to the raised issues, we have performed new experiments and made necessary changes on the manuscript.

**eLife Assessment**
The study describes a valuable new technology in the field of targeted protein degradation that allows identification of E3-ubiquitin ligases that target a protein of interest. The presented data are convincing, however, it is unclear whether the proposed system can be successfully used in high throughput applications. This technology will serve the community in the initial stages of developing targeted protein degraders.

We thank the eLife editors for the positive assessment and have clarified the scalability of our system for high throughput applications in the revised manuscript (see our response to both reviewer’s comment on weakness point 1).

**Reviewer #1 (Public Review):**
Summary:PROTACs are heterobifunctional molecules that utilize the Ubiquitin Proteasome System to selectively degrade target proteins within cells. Upon introduction to the cells, PROTACs capture the activity of the E3 ubiquitin ligases for ubiquitination of the targeted protein, leading to its subsequent degradation by the proteasome. The main benefit of PROTAC technology is that it expands the "druggable proteome" and provides numerous possibilities for therapeutic use. However, there are also some difficulties, including the one addressed in this manuscript: identifying suitable target-E3 ligase pairs for successful degradation. Currently, only a few out of about 600 E3 ligases are used to develop PROTAC compounds, which creates the need to identify other E3 ligases that could be used in PROTAC synthesis. Testing the efficacy of PROTAC compounds has been limited to empirical tests, leading to lengthy and often failure-prone processes. This manuscript addressed the need for faster and more reliable assays to identify the compatible pairs of E3 ligases-target proteins. The authors propose using the RiPA assay, which depends on rapamycin-induced dimerization of FKBP12 protein with FRB domain. The PROTAC technology is advancing rapidly, making this manuscript both timely and essential. The RiPA assay might be useful in identifying novel E3 ligases that could be utilized in PROTAC technology. Additionally, it could be used at the initial stages of PROTAC development, looking for the best E3 ligase for the specific target.The authors described an elegant assay that is scalable, easy-to-use, and applicable to a wide range of cellular models. This method allows for the quantitative validation of the degradation efficacy of a given pair of E3 ligase-target proteins, using luciferase activity as a measure. Importantly, the assay also enables the measurement of kinetics in living cells, enhancing its practicality.Strengths:(1) The authors have addressed the crucial needs that arise during PROTAC development. In the introduction, they nicely describe the advantages and disadvantages of the PROTAC technology and explain why such an assay is needed.(2) The study includes essential controls in experiments (important for generating new assay), such as using the FRB vector without E3 ligase as a negative control, testing different linkers (which may influence the efficacy of the degradation), and creating and testing K-less vectors to exclude the possibility of luciferase or FKBP12 ubiquitination instead of WDR5 (the target protein). Additionally, the position of the luc in the FKBP12 vector and the position of VHL in the FRB vector are tested. Different E3 ligases are tested using previously identified target proteins, confirming the assay's utility and accuracy.(3) The study identified a "new" E3 ligase that is suitable for PROTAC technology (FBXL).

We greatly appreciate the reviewer’s positive feedback on our work. To evaluate our system further, in our revised manuscript we have conducted additional analysis on KRASG12D degradation via VHL and CRBN within our K-less system. Consistent with previous findings of VHL-harnessing PROTACs, our assay demonstrated that VHL mediated efficient degradation of KRASG12D while CRBN induced only a minor effect. This new data is presented in Figure 2 - figure supplement 1C of the revised manuscript.

Weaknesses:· It is not clear how feasible it would be to adapt the assay for high-throughput screens.

The design of our study is a well-based assay. It is therefore possible but not realistic to evaluate all 600 and more human E3 ligases. Nonetheless, if interested in all E3 ligases, our assay could be adapted for pooled experimental strategies, as demonstrated in Poirson, J., Cho, H., Dhillon, A. et al., Nature 628, 878–886 (2024).

Our system offers several advantages over pooled screens, including the generation of more quantitative data and faster testing of selected candidates. Pooled screens, by contrast, require more time due to the necessity of next-generation sequencing and bioinformatics analysis. Moreover, in response to the reviewers comment, we have included a schematic in the revised manuscript (Figure 4 - figure supplement 1A) that outlines the assay duration and hands-on time for target and E3 ligase candidates.

· In some experiments, the efficacy of WDR5 degradation tested by immunoblotting appears to be lower than luciferase activity (e.g., Figure 2G and H).

We concur with the reviewer that in some instances, the degradation observed via immunoblotting appears lower than that indicated by luciferase activity. Thus, we have quantified the western and added it to the respective blots. This discrepancy may result from the non-linearity of western blots.

**Reviewer #2 (Public Review):**
Summary:Adhikari and colleagues developed a new technique, rapamycin-induced proximity assay (RiPA), to identify E3-ubiquitin (ub) ligases of a protein target, aiming at identifying additional E3 ligases that could be targeted for PROTAC generation or ligases that may degrade a protein target. The study is timely, as expanding the landscape of E3-ub ligases for developing targeted degraders is a primary direction in the field.Strengths:The study's strength lies in its practical application of the FRB:FKBP12 system. This system is used to identify E3-ub ligases that would degrade a target of interest, as evidenced by the reduction in luminescence upon the addition of rapamycin. This approach effectively mimics the potential action of a PROTAC.

We are delighted with this assessment of our work by the reviewer. To evaluate our system further, in our revised manuscript we have conducted additional analysis on KRASG12D degradation via VHL and CRBN within our K-less system. Consistent with previous findings of VHL-harnessing PROTACs, our assay demonstrated that VHL mediated efficient degradation of KRASG12D while CRBN induced only a minor effect. This new data is presented in Figure 2 - figure supplement 1C of the revised manuscript.

Weaknesses:(1) While the technique shows promise, its application in a discovery setting, particularly for high-throughput or unbiased E3-ub ligase identification, may pose challenges. The authors should provide more detailed insights into these potential difficulties to foster a more comprehensive understanding of RiPA's limitations.

The design of our study is well-based assay . It is therefore possible but not realistic to evaluate all 600 and more human E3 ligases. Nonetheless, if interested in all E3 ligases, our assay could be adapted for pooled experimental strategies, as demonstrated in Poirson, J., Cho, H., Dhillon, A. et al., Nature 628, 878–886 (2024).

Our system offers several advantages over pooled screens, including the generation of more quantitative data and faster testing of selected candidates. Pooled screens, by contrast, require more time due to the necessity of next-generation sequencing and bioinformatics analysis. Moreover, in response to the reviewers comment, we have included a schematic in the revised manuscript (Figure 4 - figure supplement 1A) that outlines the assay duration and hands-on time for target and E3 ligase candidates.

We also added the following sentences to the Limitations of the study section of the revised manuscript (line 322-326): “While our system offers easy testing of different tagging approaches and due to its simple workflow facilitates the rapid characterization of novel E3 ligases across multiple targets, it is currently not optimized for high-throughput evaluation of all 600+ E3 ligases. Achieving such scale would necessitate further adaptations, including the incorporation of pooled experimental strategies.”

(2) While RiPA will help identify E3 ligases, PROTAC design would still be empirical. The authors should discuss this limitation. Could the technology be applied to molecular glue generation?

We agree with the reviewer that our assay rationalizes the choice of E3 ligases but that PROTAC design (“linkerology”) is still mostly empirical. To address this, we included the following line in the Limitations of the study section of our initial manuscript (line 327-330): “Conversely, it is also conceivable that an E3 ligase that can efficiently decrease the levels of a particular target in the RiPA setting may be less suitable for PROTACs, since PROTACs that mimic the steric interaction of the target/E3 pair may not be easily identified in the chemical space.”

Regarding molecular glues, our assay could also be instrumental in identifying suitable E3 ligases for a target protein prior to screening for molecular glues, provided that the screening system specifically screens E3 ligase and target pairs. However, as most molecular glue screens are currently agnostic to specific E3 ligases or targets, our system may not be applicable in those cases. We have elaborated on this in the discussion section of the revised manuscript (line 271-274): “We envision that this setting will be valuable for identifying the most suitable E3 ligase candidates for PROTACs aimed at specific proteins, and for guiding E3 ligase selection when screening for molecular glues targeting specific E3 ligase and protein pairs.”

(3) Controls to verify the intended mechanism of action are missing, such as using a proteasome inhibitor or VHL inhibitors/siRNA to verify on-target effects. Verification of the target E3 ligase complex after rapamycin addition via orthogonal approaches, such as IP, should be considered.

We thank the reviewer for the comment. Particularly VHL siRNA is not beneficial in this setup, as we overexpress the E3 ligase rather than relying on endogenous protein.

To verify mechanism of action, we performed additional experiments in the presence of proteosomal inhibitor MG132 and neddylation inhibitor MLN4924 with target KRASG12D and E3 ligase VHL. The results is shown in Figure 2H of the revised manuscript.

Minor concern:The graphs in Figure 1E are missing.

We thank the reviewer for pointing this out. We corrected the figure in the revised manuscript.

**Reviewer #1 (Recommendations For The Authors):**
• Optionally, the authors could add control experiments with Aurora B and Crb vectors (there shouldn't be any degradation) and experiments confirming that the degradation occurs via the proteasome. For example, the addition of proteasome inhibitors (such as bortezomib) should decrease the efficiency of the target degradation and confirm that targets are degraded via the proteasome system.

Regarding Aurora-B degradation, as far as we know, there are no specific Aurora-B PROTACs reported. Thus, there is no definitive evidence that CRBN could not degrade Aurora-B. Nevertheless, we performed assays with Aurora-B and VHL, CRBN, or FRB, and observed more effective degradation of Aurora-B by VHL than CRBN. This data is now included in Figure 2 - figure supplement 1B of the revised manuscript.

• It would also be helpful to provide a possible explanation for why the ratio 1:1 of vectors did not induce the degradation (regarding Figure 1D).

We believe the lack of degradation with 1:1 vector ratio is due to the differential expression levels of endogenous FKBP12 and mTOR in HEK293 cells. According to Human Protein Atlas, the normalized protein-coding transcripts per million (nTPM) for FKBP12 and mTOR in HEK293 cells are 160 and 24 respectively, indicating that FKBP12 is expressed at levels approximately 6.7 times higher than mTOR. This disparity likely limits the heterodimerization of exclusively fusion proteins upon rapamycin addition. To increase the likelihood of FKBP12 and FRB fusion protein dimerization, we used a higher ratio of the FRB component during transfection, considering the higher endogenous expression of FKBP12.

• It would be helpful to add more explanation for the data in Figure 1F, including whether there is a difference between vectors with different positions of VHL and FRB and why the FRB-VHL vector is less expressed without rapamycin.

We thank the reviewer for the comment. Regarding the vector orientations of VHL/FRB and WDR5/Luc/FKBP12, we have consistently observed different migration behaviors for WDR5 and VHL constructs, despite their same molecular weights. This observation aligns with literature reports where differential running behavior is noted when FRB or FKBP12 (or their mutants) are tagged to the N- or C-terminus of a protein (Bondeson, D.P., Mullin-Bernstein, Z., Oliver, S. et al. Nat Commun 13, 5495 (2022); Mabe, S., Nagamune, T. & Kawahara, M. Sci Rep 4, 6127 (2014)). We have now included the following explanation in the figure legend of Figure 1F of the revised manuscript: “WDR5 and VHL fusion proteins tagged at the N- and C-terminal show different migration behaviors despite having same molecular weight.”

Additionally, the stabilizing effect of rapamycin on FRB (or its mutants), FRB fusion proteins, and FRB-containing proteins has been documented (Stankunas, K., Bayle, J.H., Havranek, J.J. et al. ChemBioChem, 8(10), 1162-1169 (2007); Stankunas, K., Bayle, J.H., Gestwicki J.E. et al. Mol Cell, 12(6), 1615–1624 (2003); Zhang, C., Cui, M., Cui, Y. et al. J. Vis. Exp. (150), e59656 (2019)). We believe that the degree of stabilization by rapamycin could differ between N- and C-terminal FRB fusion proteins.

• Finally, the mistake in Figure 2G (where the lanes are wrongly labelled, BRBN-FRB and FRB) should be corrected. Also please correct the graph in Figure 1E (there seems to be a problem with bars for 1:100). There are some typos, such as in lines 38, 277, and 288.

Thank you for bringing this to our attention. We have corrected all the mentioned errors.